# Molecular Dynamics Simulation Study of the Self-Assembly of Phenylalanine Peptide Nanotubes

**DOI:** 10.3390/nano12050861

**Published:** 2022-03-03

**Authors:** Vladimir Bystrov, Ilya Likhachev, Alla Sidorova, Sergey Filippov, Aleksey Lutsenko, Denis Shpigun, Ekaterina Belova

**Affiliations:** 1Institute of Mathematical Problems of Biology—Branch of Keldysh Institute of Applied Mathematics, RAS, 142290 Pushchino, Russia; ilya_lihachev@mail.ru (I.L.); fsv141@mail.ru (S.F.); 2Faculty of Physics, Lomonosov Moscow State University, 119991 Moscow, Russia; sky314bone@mail.ru (A.S.); aleksluchrus@yandex.ru (A.L.); denish.den@mail.ru (D.S.); ev.malyshko@physics.msu.ru (E.B.)

**Keywords:** molecular dynamics method, MD manipulator, controlled molecular dynamics, self-assembly of nanostructures, nanotubes, phenylalanine, chirality

## Abstract

In this paper, we propose and use a new approach for a relatively simple technique for conducting MD simulation (MDS) of various molecular nanostructures, determining the trajectory of the MD run and forming the final structure using external force actions. A molecular dynamics manipulator (MD manipulator) is a controlled MDS type. As an example, the applicability of the developed algorithm for assembling peptide nanotubes (PNT) from linear phenylalanine (F or Phe) chains of different chirality is presented. The most adequate regimes for the formation of nanotubes of right chirality D from the initial L-F and nanotubes of left chirality L of their initial dipeptides D-F modes were determined. We use the method of a mixed (vector–scalar) product of the vectors of the sequence of dipole moments of phenylalanine molecules located along the nanotube helix to calculate the magnitude and sign of chirality of self-assembled helical phenylalanine nanotubes, which shows the validity of the proposed approach. As result, all data obtained correspond to the regularity of the chirality sign change of the molecular structures with a hierarchical complication of their organization.

## 1. Introduction

The development and creation of new computer technologies [1,2], including those in the field of molecular dynamics (MD) simulation (MDS) methods [3,4,5,6,7], is a promising and practically important modern scientific trend. It is also significant and promising for studying the fundamental processes of self-organization and self-assembly of molecular structures [8,9,10].

Self-assembly of amino acids and short peptides is the basis for the formation of many complex biomolecular structures and systems in biology and medicine, as well as in nanobiotechnology and contemporary nanomaterials design [2,8,10]. As a result of such self-assembly, they usually form helix-like or more complex self-organizing structures of different levels of their hierarchical organization [11,12,13]. At the same time, chirality plays an important role in the processes of self-assembly of biomacromolecules, both in their initial structures and in various complicated structures at subsequent levels of their hierarchical organization [14,15,16,17,18].

With the complication of self-organization of macromolecules, the sign of chirality alternates: for amino acids and proteins, starting from L to D. That means, from the “left” (L—from the Latin “laeva”) to the “right” (D—from the Latin “dextra”) enantiomer. These two forms of enantiomers (or optical isomers) rotate the plane of polarized light either to the right (D is the right screw) or to the left (L is the left screw). It creates the L-D-L-D sequence for proteins. While for the DNA-based replication system, the sequence is different—D-L-D-L, “conjugated” to peptide and protein molecules [15,16,17]. The alternation of the sign of chirality upon transition to a higher hierarchical level turns out to be a necessary factor [15,16,17,18].

Studies of the processes of self-organization and self-assembly of various complicated (including helix-like) structures based on amino acids have been intensively carried out in recent years [8,12,13,18]. Among the approaches used, the methods of molecular dynamics (MD) are probably of the greatest interest [3,4,5,6,7,10].

In this paper, we consider one of these new approaches, namely the so-called molecular dynamics (MD) manipulator, which allows efficient assembly of simulated molecular structures, including taking into account the chirality of molecular components, using external force actions. It is a controlled MDS type—a molecular dynamics manipulator (MD manipulator). It is an imitation of the operation of a device by applying force to the existing initial structure to obtain a new final structure having the same chemical composition but a different geometry (topology). The PUMA-CUDA (Pushchino Molecular Analysis (PUMA) with support of Compute Unified Device Architecture (CUDA)) software package was used as the main MD modeling program, which uses the physics of the PUMA software package, developed in the IMPB molecular dynamics laboratory headed by N.K. Balabaev [19,20]. Using this MDS tool, one can investigate the formation of helix-like structures from a linear sequence of any amino acid variation.

As an example, in this work, to demonstrate the process of self-organization and self-assembly of complex biomolecular structures using the developed algorithm of the MD manipulator, the formation of a helix-like structure of the peptide nanotube (PNT) type based on linear chains of phenylalanine amino acids (F or Phe) of different chirality (left L-Phe and right D-Phe) [12,17,21] is shown.

It is known that diphenylalanine dipeptides (FF or (Phe)_2_) are also formed from such a phenylalanine amino acid (F or Phe) and are assembled into similar peptide nanotubes [12,17,22,23,24,25,26,27,28,29,30,31,32,33,34,35,36,37,38,39,40,41,42,43,44,45,46,47,48,49,50]. Diphenylalanine dipeptide FF and peptide nanotubes (PNTs) based on it (FF PNTs) are currently quite well studied, as they are of considerable interest due to their special structural and physical properties important in various applications [17,44,45,46,47]. These self-assembled FF PNTs are biocompatible [12,22,23,24,25,26,27,28,29,30,31,32] and exhibit excellent mechanical [33,34], chemical and thermal stability [35] in biomedical applications [12,17,22,23,24,25,26,27,28,29], and they have interesting electronic [36], optical [37,38,39,40] and pyroelectric [41] properties, as well as strong polarization [12,17,43,44,45,46,47,48,49,50] and exceptional piezoelectric effect [12,42,43,44,45,46,47], making them promising functional material candidates for various sensors and micro/nano-electronic devices.

Self-assembly of diphenylalanine FF has long been known. Diphenylalanine is a key motif that forms amyloid fibrils of the Aβ-peptide involved in Alzheimer’s disease [17,25,27,28,29]. Single phenylalanine F molecules can also form nanofibrils (or nanotubes) [51,52]. As shown in [52], synthesized fibrils (similar to the peptide nanotubes), as well as those obtained in experiments from mice with phenylketonuria, also have amyloid-like characteristics.

In this paper, we consider modeling the self-organization of a helical structure such as a peptide nanotube based on phenylalanine molecules. To perform such self-organization on a model from a set of 48 molecules of phenylalanine, (Phe)_48_ (or F_48_), into a spiral tubular structure (F PNT), an algorithm for assembling a phenylalanine nanotube uses one of the approaches of the molecular dynamics method—the MD manipulator [21]. The MD manipulator method is a controlled MDS self-assembly process that makes it possible to obtain a new structure having the same chemical composition but different geometry and topology by applying special forces to the existing original structure.

Thus, the structure of a spiral peptide nanotube (F PNT) is formed from a linear chain of 48 molecules (Phe)_48_ using the MD manipulator method. At the same time, depending on the chirality of the initial molecule F, the chirality of the self-organizing helix-like structure has a different sign in accordance with the established regularity of the changing of chirality sign with the complication of the hierarchical molecular structures [12,15,16,17,18].

Purposeful and interactive work of the MD manipulator will allow determining the most adequate regimes for the formation of a nanotube of the “right” chirality D from the initial “left” amino acid L-F of phenylalanine and a nanotube of the “left” chirality L from the initial “right” molecule D-F.

Another important task of this work is to calculate the magnitude and sign of the chirality of the self-assembled helical structures of phenylalanine nanotubes. For this purpose, we will apply the method based on the calculation of the mixed vector–scalar product of the vectors of dipole moments of individual phenylalanine molecules located sequentially along the helix line of phenylalanine nanotubes [53,54]. This algorithm for chirality calculation based on dipole moment vectors is discussed in detail in the second part of this article.

## 2. Model Details and Methodology for Numerical MD Experiments

### 2.1. Main Software Platform Used

The previously developed PUMA-CUDA software package, which uses the physics of the PUMA software package [19,20] and is capable of performing calculations in a parallel mode both on personal computers, including those with graphics accelerators, and on high-performance heterogeneous computing clusters, was used as the basis for the molecular dynamics simulation program. One of the authors is a developer of this package. It is the ability to easily make changes to the software package that makes the molecular dynamics simulation program an MD manipulator [21,55,56]. We can also change the parameters of the force field during the MD experiment in the interactive mode.

Further, the program has added the ability to create additional force effects in the form of Hooke’s “springs” with a given stiffness and nominal length. Springs can connect not only any two chosen and indicated atoms. For regular (repeating structures) it is possible to indicate groups of pairs of atoms by determining the numbers of initial atoms and their periodicity. All work of this program is carried out interactively. In this case, two programs are used on a personal computer: the PUMA-CUDA program directly conducts the simulation itself, and the additional Trajectory Analyzer of Molecular Dynamics (TAMD) program [21,57,58] connects to the PUMA-CUDA program using client–server technology and visualizes the current state of the system in real time. In this mode, it is convenient to set various force actions [59] in the course of a numerical experiment (for example, set or change the stiffness of springs) and then observe how this affects the system.

### 2.2. Model Details and Force Technique Assembly

To build a model in this work, a ready-made file with the coordinates of all atoms of a chain of the 48 phenylalanine (Phe or F) molecules stretched in a row—Poly(Phe)_48_—is taken as a basis, which has the standard PDB format. It can be prepared using a molecular chemistry constructor such as HyperChem [60] or PyMol (see Figure 1). Here are shown fragments of chains of 12 Phe molecules, of different chirality L and D (L-F and D-F, or L-Phe and D-Phe); each Phe (or F) molecule has 20 atoms. Therefore, the total chain has 12 × 20 = 240 atoms.

In this MD simulation, 48 Phe molecules are used, that is, a total of 20 × 48 = 960 atoms. In the process of MD simulation, it is required to obtain a helix with a step of 12 amino acids, per one turn of the helix, i.e., the step is 12 × 20 = 240 atoms per one turn (or coil). As a result, four turns of 12 Phe (F) molecules are formed in total. We fix the first atom to avoid displacement of the molecule in space. The PUMA-CUDA software package allows us to fix the coordinates of the required numbers of atoms. To run a controlled MD simulation, we add two types of springs to the force field.

The first type of springs provides a diametrical stretch; the second type provides a longitudinal bond along the nanotube axis. For the diametric stretching, we need to obtain an inner diameter of the size ~12 Å, and for a cylindrical helix bond, we need to obtain a helix pitch of the size ~5.44 Å (as follows from the experimental data). To do this, we connect the second atom with a spring to the atom 2 + 20 × 6 = 122. The length of the spring is 12 Å. Then we connect the 22nd atom with the 142nd, and so on. We obtain 41 springs of diametrical extensions along the entire length of the spiral (see Figure 2a).

The second type of springs will provide a cylindrical helix bond, the helix pitch. That is, we connect the first amino acid with 13, the second with 14, the third with 15, etc. The length of the spring would have to be 5.44 Å, similar to [45,46,47,48,49,50] (see Figure 2b). However, to forcefully attract several amino acids, it is advisable to use a smaller spring. In our case, this is 3 Å. Accuracy is not important here. In place of 3 Å there could be another number, up to zero. It is only important to indicate the direction of the force.

The stiffness of the spring is gradually increased. After 90 ps of this MD simulation, the linear structure is transformed into a helix-like nanotube shape (see Figure 3a). After that, we add springs that provide additional diametrical stretching of amino acid residues (force is applied to the terminal hydrogen atom). It is necessary to obtain a clear structure so that all residues (more precisely, aromatic rings of all amino acid residues) are located on the outside (Figure 3b). The nominal length corresponds to the initial dimensions (radius and pitch) of the resulting helix minus 1–2 Å. The subtraction is done to ensure the contraction of the structure towards the original positions. The spring constant changed 10 times from 0.00001 to 1 pN/Å every 10 ps. AMBER99 [59] was used as the force field. Cut-off radius is 10.5 Å. Numerical integration step is 0.001 ps. Temperature is 300 K. We used a collisional thermostat [19,20]. The barostat was not used because the simulation was carried out in vacuum.

It is possible to obtain a similar structure from a larger number of initial F molecules in a linear chain corresponding to its chirality (Figure 4).

Thus, as a result of MD modeling of the self-assembly process, having monomers of different initial chirality LF (L-Phe) and DF (D-Phe) under the same external influences in the process of self-assembly by the MD manipulation method with a certain and fairly high probability, we obtain spiral nanotubes of the corresponding chirality: right-handed (D) from L-monomers and left-handed (L) from D-monomers.

## 3. Results of MD Analysis and Discussion of Results

Thus, here we have demonstrated a technique for constructing a nanotube from a linear Poly(Phe)_48_ structure using additional force effects implemented in the PUMA-CUDA molecular dynamics simulation program [19,20,55,56]. The work was carried out interactively using a client–server bundle of molecular dynamics simulation and trajectory display programs (Trajectory Analyzer of Molecular Dynamics (TAMD)) [57,58].

To test the statistical reliability of this method, 32 implementations of the assembly of a phenylalanine nanotube by the described MD manipulator from linear monomers of chirality L and D were carried out. The relaxation time and force effects in all implementations are the same. However, all implementations of computational experiments are statistically independent due to the use of different sequences of the random number generator, which is used in the collisional thermostat [55,56].

For a large number of realizations, not all monomers of chirality D yield left-handed helices, but L-monomers yield right-handed ones. The list of received spirals is given in Table 1 [21].

The fractional numbers in Table 1 indicate the simultaneous twisting of the monomer from one end to the left helix and from the other to the right. These phenomena can be explained by “distortions” arising from the limited time of the computational experiment. The probability of obtaining a helix of the desired chirality depends on the amount of time allotted for the steps described in the nanotube assembly technique.

With a total number of realizations of MD runs N = 32, we obtain for the right D-Phe monomer the number of left-handed helices N_D-L_ = 26.2. Accordingly, we can estimate the probability of a self-assembly P_D-L_ = N_D-L_/N = 26.2/32 = 0.81875 ≈ 82%. Similarly, for left L-Phe monomers, we obtain, with the frequency of occurrence of right-handed helices on their base N_L-D_ = 28.6, the final probability of assembly of right-handed helices P_L-D_ = N_L-D_/N = 28.6/32 = 0.89375 ≈ 89% [21].

These results are also similar to self-organization and change of sign of chirality during self-assembly of peptide nanotubes based on diphenylalanine with different chirality [46,47,48,49,50].

Thus, in this work, we developed a method for constructing helical nanotubes from the linear structure Poly(Phe)_48_ of various chirality L-Phe and D-Phe using additional force actions implemented in the PUMA-CUDA molecular dynamics simulation program [19,20,21,55,56,57,58]. The statistics of self-assembly of nanotubes into helical structures of different chirality were obtained depending on the specified modes of the MD simulator. The results obtained were analyzed in comparison with the data of other works on modeling such nanotubes of various chirality and experimental data.

As a result, the regimes most adequate for the formation of nanotubes of right chirality D from initial L-Phe peptides and nanotubes of left chirality L from D-Phe peptides were determined. These data fully correspond to the established law of molecular structure chirality sign change with hierarchical complication of the level of organization [12,14,15,16,17,18].

Thus, as a result of the performed MD runs and calculations, we obtain two types of helix-like nanotubes with a different screw or different chirality: a nanotube with right-handed D chirality (D-PNT_L-F48) based on 48 phenylalanine molecules with initially opposite L chirality (L-F48) and a left-chirality L nanotube (L-PNT_D-F48) based on 48 phenylalanine molecules with initially opposite D chirality (D-F48). For them, the corresponding pdb files were also obtained with the coordinates of all 48 atoms (Figure 5), which can be used for further research.

In this work, using the data of the obtained pdb files, we will calculate the magnitude and sign of the chirality of these self-assembled nanotubes. This novel method is based on the values of the dipole moments of each amino acid or peptide [53,54]. Therefore, we need to obtain the values of the dipole moments of individual phenylalanine molecules in the coils of these nanotube helices.

## 4. Calculation of the Chirality of Phenylalanine Helical Nanotubes from the Dipole Moment Data of Their Constituent Phenylalanine Molecules

A method for calculating the chirality magnitude and sign of helix-like peptide nanotubes (PNT) using the vectors of the successive dipole moments of their constituent peptide molecules was proposed and developed in works [53,54]. It is based on the procedure of the mixed product of three consecutive vectors in a coil of a helix-like structure.

In this work, using the above obtained helical structures of phenylalanine nanotubes of different chirality, we calculate the magnitude and sign of their chirality by this method.

As is well known [9,10,11,12,13], all amino acids have dipole moments due to their structural arrangement, mainly due to the difference in charges (and their signs) between the amine and carboxyl groups. In molecules such as phenylalanine, of course, aromatic rings also play a role, but since they are generally neutral, they play mainly the inertial role of shifting the center of mass. It is also well known that the dipole moment is the vector **D**, which has three components in Cartesian coordinates: *D_x_, D_y_* and *D_z_*.

Quantum mechanical methods are used to calculate the dipole moments of various atomic and molecular structures, including amino acids. Quite convenient and fast quantum calculation methods here are semiempirical quantum-chemical methods [60,61,62,63,64,65,66,67,68,69,70,71,72,73]: AM1, RM1, PM3, PM6, PM7, etc.

Many of them are implemented in HyperChem [60] and MOPAC [65,66] programs. More developed ones are implemented in MOPAC2016 [66], such as PM7 and PM6-D3H4X [67].

Here, it is also important to take into account the chirality of the studied amino acid molecules. The absolute values of the dipole moments of the phenylalanine molecules considered here for cases of different chirality L-F and D-F are close to each other, but the orientation of their vectors **D** is different and is determined by the different structural organization of these molecules in different chiral isomers (Figure 6).

Table 2 shows the results of calculations of the initial dipole moment values for L-F and D-F of individual free phenylalanine F molecules by different quantum semiempirical methods.

From comparing their values and analyzing the accuracy by different calculation methods (taking into account also the data from [67,68]), we can conclude that, for example, the PM3 method is not so reliable and correct here, while the PM7 and RM1 methods are quite close to each other in the terms of final results. Therefore, we further use the RM1 method in specific calculations.

In the helix-like structures of the different chirality D-PNT and L-PNT obtained above, there are 48 molecules of the L-F and D-F chirality, correspondingly (each with its dipole moment **D** and its orientation in the space, represented by three components *D_x_*, *D_y_* and *D_z_*). These values of the dipole moment vector **D** for F molecules are formed as a result of the self-consistent interaction of individual dipoles in the helical structure of the entire nanotube, and they differ in magnitude and their components from isolated dipoles **D** for F molecules.

To perform the necessary calculations, we will select one helix coil (or turn) from each helical nanotube of the different chirality, calculate all their dipole moments and then apply our calculation method to them, based on the mixed product of three dipole moment vectors from a number of the successive phenylalanine molecules that form this helix coil (turn) of phenylalanine peptide nanotubes (F PNTs) with consideration of their chirality.

Helix-like PNT structures (obtained above) based on phenylalanine of the different chirality (L and D), consisting of 48 F molecules and containing 4 coils in such structures of nanotubes of each chirality L-F48 and D-F48 (with coordinates of all atoms in standard *.pdb format), are shown on Figure 5. For the analysis of the dipole moments, we transfer them from *.pdb format files to the HyperChem [60] workspace (into *.hin format with Cartesian x, y, z coordinates for all atoms), which are shown in Figure 7.

Further, we select one coil at a time from each PNT helix, consisting of four coils, and calculate their common dipole moment for each coil with different chirality. When calculating the dipole moments of these coils, they have opposite directions of the vectors of the total dipole moments **D_L-F_** for each L-F and **D_D-F_** for each D-F coil (Figure 8a,b).

A further step of these calculations is a selection of each consequential phenylalanine F molecule from the corresponding coil, containing 12 F molecules, and the calculation of its dipole moment ***D_i_*** using various methods (from the MOPAC program [66] and HyperChem package [60,68]): quantum-chemical semiempirical methods PM7 and RM1. This procedure is schematically shown in Figure 9.

The origin of each **D***_i_* vector is taken relative to the center of mass of the corresponding F molecules. The absolute value of each dipole moment **D***_i_* is
(1)$$D_{i}=\left|D_{i}\right|=\sqrt{D_{x,i}^{2}+D_{y,i}^{2}+D_{z,i}^{2}},$$
where *D_x,i_, D_y,i_* and *D_z,i_* are the components of the *i*-th vector **D*_i_*** in the Cartesian coordinates.

Using these vectors of dipole moments, located in series along the line of the nanotube helix, we construct and calculate the value of their mixed (vector–scalar) product. Recently, in the works [74,75] such a method was proposed and developed for determining the chirality of protein structures by the mixed (vector–scalar) product of vectors. As noted in these works, to estimate the sign of chirality of protein structures, a sufficient condition is the mutual arrangement of α-carbon atoms—reference points in helices, turns and loops of protein molecules. This makes it possible to build successive vectors connecting these reference points in protein helices and then, using the mixed vector product method, calculate the magnitude and sign of the chirality of helix-like structures. In the works [53,54] this approach was developed and applied to the sequence of dipole moment vectors of individual phenylalanine and diphenylalanine molecules located along the helix line of such a peptide nanotube. Therefore, according to works [53,54], here the sum of the mixed (vector–scalar) products of the dipole moments related to the PNT’s chirality can be written as follows:(2)$$c_{total}=\overset{n-2}{\underset{i=1}{\sum }}\left(\left[D_{i},D_{i+1}\right],D_{i+2}\right),$$

It is necessary to note that the summation here is taken over *i* in the range from 1 to *n* − 2, and *n* = 12. The *c_total_* can be normalized over the cube of the average value of the total dipole momentum of the PNT’s coil, $D_{av}=\frac{1}{12}\overset{12}{\underset{i=1}{\sum }}D_{i}$, to obtain a universal measure of the chirality:(3)$$c_{norm}=\frac{c_{total}}{D_{av}^{3}}.$$

The results obtained here as well as earlier in [53,54] show a characteristic change in the sign of chirality during the transition to a higher level of organization, which is observed in the structures of biomacromolecules [15,16,17]: the calculated chirality of a helix-like nanotube based on the L-F initial amino acid has a positive sign, D type, and the chirality of the D-F based nanotube has a negative sign corresponding to the L chirality type (see Table 3).

Note that the data presented and used here for the cube of average values of the absolute value of the total dipole moment $D_{av}=\frac{1}{12}\overset{12}{\underset{i=1}{\sum }}D_{i}$ of each of the F amino acids represent the average volume built on three consecutive vectors of the mixed product of these vectors. For each group of three such vectors, the calculated value of the magnitude of their mixed product changes, which is the corresponding volume on these three vectors. Normalizing here on the average value (*D_av_*)^3^ according to the Equation (3), we obtain the relative change in this volume, slightly changing around the absolute value of “1”. Moreover, for the left-handed and right-handed triplets of vectors, the corresponding volumes have different signs, which leads to a change in sign in this case.

## 5. Conclusions

The obtained results of calculating the magnitude and sign of chirality for L- and D-nanotubes based on phenylalanine (Table 3) are similar to the data of other works on modeling peptide and dipeptide nanotubes of different chirality and experimental data [21,53,54]. These data also fully comply with the regularity of the change in the chirality sign of molecular structures with the complication of their hierarchical level of organization [15,16,17]. Therefore, this method for calculating the magnitude and sign of chirality based on the mixed product of vectors method earlier proposed by Sidorova et al. [74,75], using the values of the dipole moment vectors in the sequence of individual peptides and dipeptides (or amino acids) as vectors, is also quite adequate and can be successfully applied for assessment of the magnitude and sign of chirality of complex self-organizing helix-like nanostructures based on amino acids and various peptides/dipeptides.

The above-obtained MD simulation results of the phenylalanine helix-like nano-tubes are in good agreement with experimental observations and provide important molecular information for further research. It is interesting to note that the here-proposed molecular dynamics manipulator (MD manipulator) is a controlled type of MDS with an external force field, and in this, it is similar to the so-called steerable molecular dynamics (SMD) simulation, designed for forced protein unfolding and protein–protein separation [76,77,78,79]. In this SMD method, usually, one point of the protein is fixed and a force field is applied to the protein–protein complex to unfold or untangle it.

In our PUMA-CUDA program, there is the possibility of a similar forceful unfolding of molecules, and we conducted such numerical experiments [55,80,81]. However, it should be noted that the reverse experiment—for folding—is much more time-consuming algorithmically. Moreover, it will not always be successful. The system can slide into a local minimum of potential energy, which will prevent it from taking the required configuration. So, these approaches need special further study. However, it is interesting and promising that there is a connection between such various MDS methods.

## Figures and Tables

**Figure 1 nanomaterials-12-00861-f001:**
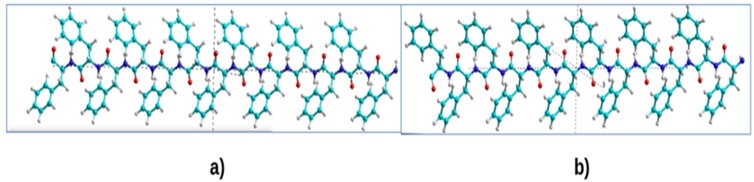
The initial structure of Poly(Phe)_48_. Models were constructed in the HyperChem program [32]: (**a**) L-Phe (L-F) β-sheet; (**b**) D-Phe (D-F) **β**-sheet. Reprinted with permission from [21].

**Figure 2 nanomaterials-12-00861-f002:**
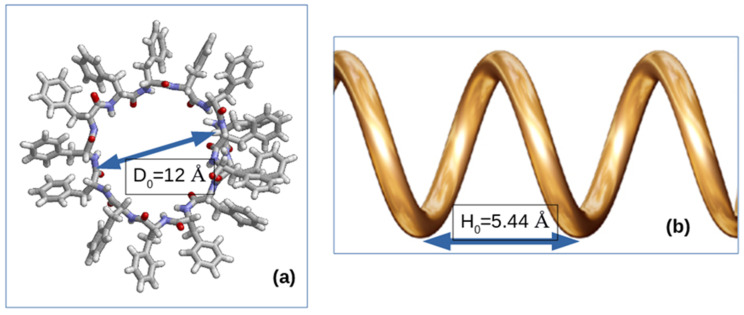
Additional force impacts: (**a**) “diametric stretch marks; (**b**) “helix pitch”. Reprinted with permission from [21].

**Figure 3 nanomaterials-12-00861-f003:**
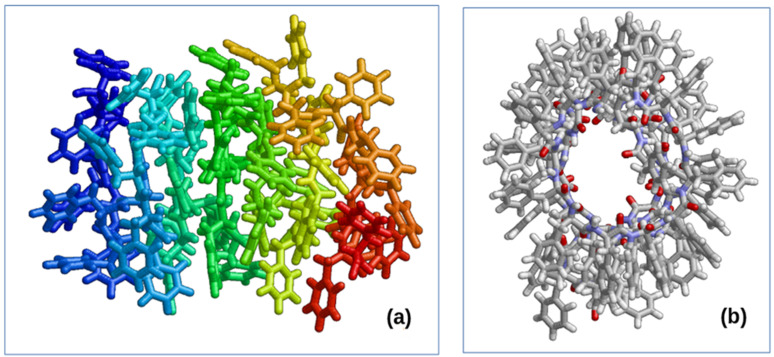
Final results: (**a**) after giving a helix-like shape; (**b**) after adding additional diametric springs, pulling back amino acid residues. Reprinted with permission from [21].

**Figure 4 nanomaterials-12-00861-f004:**
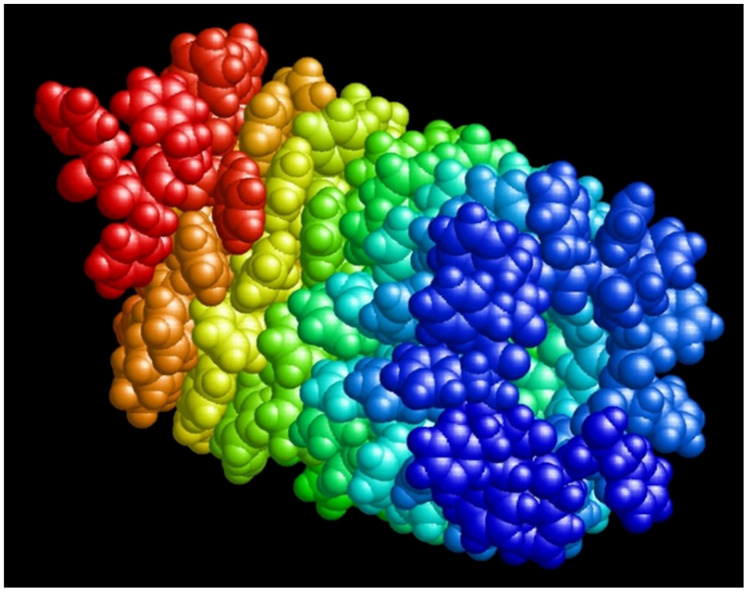
Phe nanotube of 100 amino acid residues. Reprinted with permission from [21].

**Figure 5 nanomaterials-12-00861-f005:**
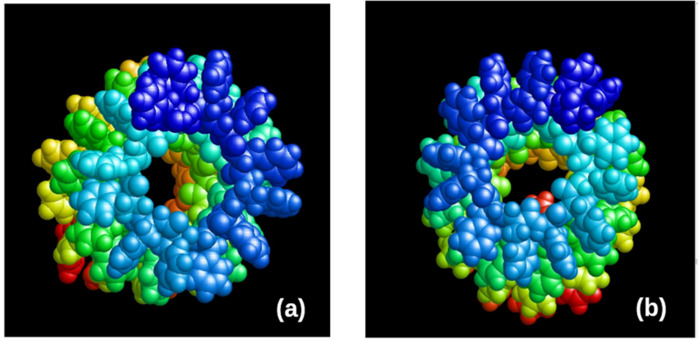
The final results obtained by MDS and self-assembly of phenylalanine helix-like structures: (**a**) a right-handed chiral D-PNT nanotube based on an MD assembly of 48 phenylalanine molecules L-F48 of the initial left-handed chirality L-F or L-Phe (corresponding pdb file L-F48.pdb); (**b**) left-handed chiral nanotube L-PNT based on MD assembly 48 phenylalanine molecules D-F48 of the initial right-handed chirality D-F or D-Phe (corresponding pdb file D-F48.pdb). Reprinted with permission from [54].

**Figure 6 nanomaterials-12-00861-f006:**
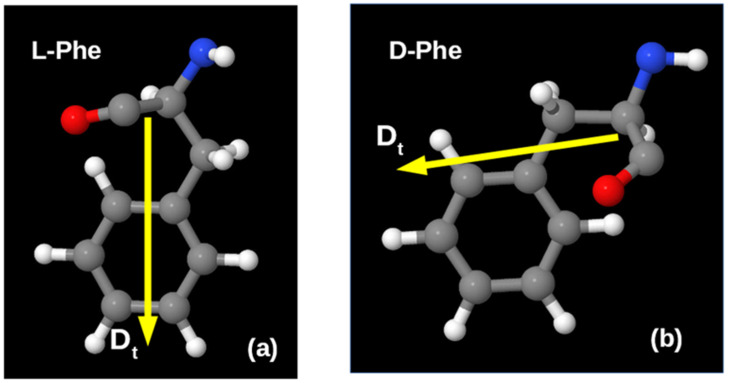
Shape of phenylalanine molecules and total dipole moment **D_t_ = D** orientation: (**a**) for the left-handed chiral isomer L-F (or L-Phe); (**b**) for the right-handed chiral isomer D-F (or D-Phe).

**Figure 7 nanomaterials-12-00861-f007:**
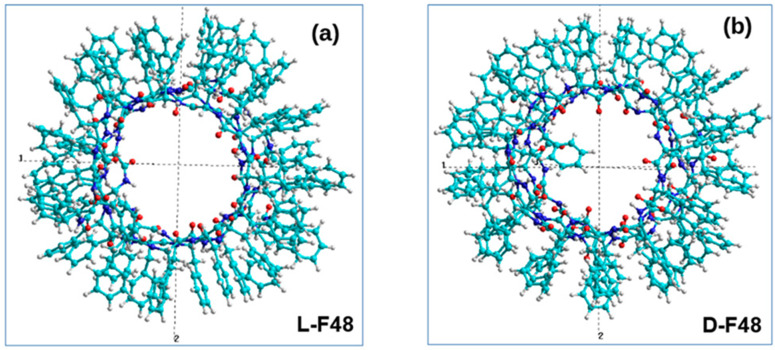
The obtained results of MDS and self-assembly of phenylalanine helix-like PNTs, transferred into HyperChem workspace (in Z-projection): (**a**) L-F48; (**b**) D-F48 (Reprinted with permission from [54]).

**Figure 8 nanomaterials-12-00861-f008:**
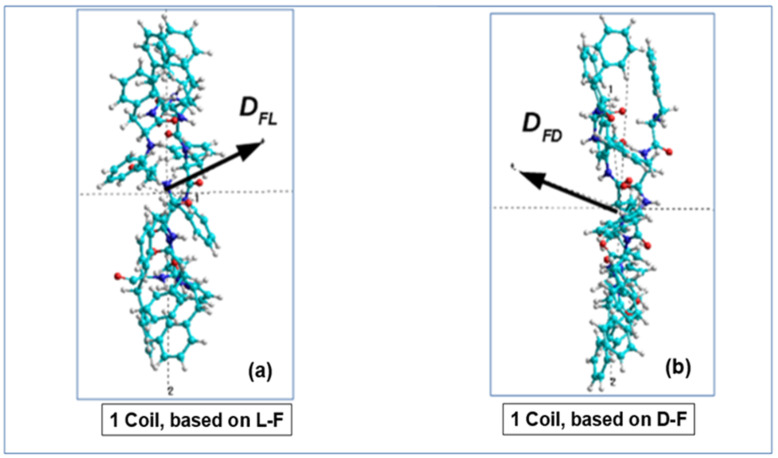
Images of one coil selected and cut from the four coils of the PNT helix-like structures (in X-projection): (**a**) for a PNT coil based on L-F; (**b**) for a PNT coil based on D-F (Reprinted with permission from [54]).

**Figure 9 nanomaterials-12-00861-f009:**
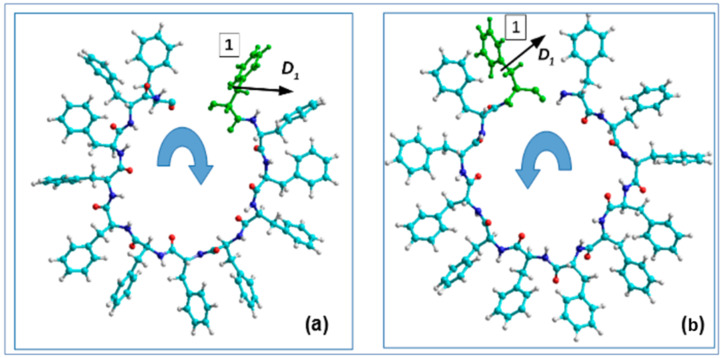
Scheme of the selection procedure of each consequential phenylalanine F molecule (from 1 to 12) from one corresponding coil of PNT helix and calculation of its dipole moment **D_i_** (for i = 1, 12) (in Z-projections): (**a**) for the D-PNT based on phenylalanine of left-handedness L-F; (**b**) the same for L-PNT based on right-handedness D-F, correspondingly. (Reprinted with permission from [54]).

**Table 1 nanomaterials-12-00861-t001:** Statistics of the helix-like structures were obtained using the MD manipulator technique (Reprinted with permission from [21]).

Implementation Number	D-Monomer		L-Monomer	
	Number of Left Helices	Number of Right Helices	Number of Left Helices	Number of Right Helices
1	0.5	0.5		1
2	0.5	0.5		1
3	1			1
4	1			1
5	1		0.2	0.8
6	1		0.5	0.5
7	1		1	
8	0.5	0.5	1	
9	1			1
10	1		0.2	0.8
11	0.2	0.8		1
12	1			1
13	1			1
14		1		1
15	1			1
16	1			1
17	1			1
18	1			1
19	1			1
20	1			1
21	0.5	0.5	0.2	0.8
22	0.5	0.5		1
23	1			1
24	1			1
25	1			1
26	1			1
27	1		0.1	0.9
28	0.5	0.5		1
29	1			1
30	0.5	0.5		1
31	0.5	0.5		1
32	1		0.2	0.8
**Total**	**26.2**	**5.8**	**3.4**	**28.6**

**Table 2 nanomaterials-12-00861-t002:** The absolute value of the phenylalanine total dipole moment computed by various methods (all data in debyes; SP/1SCF is single point (SP) calculations in HyperChem [60] or the keyword 1SCF is used in MOPAC [66] calculations, when no local optimization of the energy of the F molecule is performed; opt—the geometrically optimized F molecule performed).

#	Method	L-F		D-F	
		SP/1SCF	Opt	SP/1SC	Opt
1	PM3	1.978	1.981	1.831	1.512
2	PM7	3.167	2.684	2.756	3.323
3	PM6-D3H4	2.973	2.578	2.608	2.374
4	RM1	2.470	2.172	2.252	4.123

**Table 3 nanomaterials-12-00861-t003:** Magnitudes and signs of the chirality values obtained for L-F and D-F PNTs.

Initial Chirality Type of F Molecule	L-F	D-F
Calculating method	RM1 RHF	RM1 RHF
Computed data *c_total_,* debye^3^	20.266	−19.647
Normalized data *c_norm_*	1.219	−0.674
Chirality PNT sign	Positive	Negative
Chirality PNT symbol	**D**	**L**

## Data Availability

The data presented in this study are available on request from the corresponding author.

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
