# Peer review of "Molecular Dynamics Simulation Study of the Self-Assembly of Phenylalanine Peptide Nanotubes"

_nanomaterials, 2022, doi:10.3390/nano12050861_

Round 1

Author Response

Dear Reviewer # 1 !

Thank you very much for your comments.

All replies are in attached file. 

Reviewer 2 Report

 Molecular Dynamics Simulation Study of the Phenylalanine Peptide Nanotubes Self-Assembly

While the work could be interesting to some readers of the journal, it is not a well written ms. There are issues specially in writing, and in explaining the fundamental physics. The authors the work have tried their best to report results obtained from modeling, yet the paper lacks interpretation. Otherwise, the ms is equipped with good number of pictures, with the results obtained from some packages like PUMA, HyperChem or PyMol.

That the paper is not well written can be understood after reading the following paragraph. The first sentence is too long and is grammatically incorrect.

The PUMA-CUDA software package, which uses the physics of the PUMA software 112 package [9, 10], developed earlier and capable of performing calculations in parallel mode 113 both on personal computers, including those with graphics accelerators, was used as the basis for the molecular dynamics simulation program. and on high-performance heterogeneous computing clusters. It is the ability to easily make changes to the software package (directly during the MD run of the simulated structures in the interactive mode) that makes the molecular dynamics simulation program the so-called MD constructor [11, 29, 30].

I urge the authors to check the paper with a native English writer, as well as add some physics and chemistry to strengthen the quality of the hard work. Presenting MD simulation techniques do not constitute a paper in nanomaterials, even though the molecular frameworks considered are certainly of nanoscale entities.

Author Response

Dear Reviewer # 2 !

Thank you very much for your comments.

All replies are in attached file. 

Reviewer 3 Report

The paper presents research on molecular dynamics simulation study of the phenylalanine peptide nanotubes self-assembly. The current form's presentation of methods and scientific results is unsatisfactory for publication in the Nanomaterials journal. The minor and significant drawbacks to be addressed can be specified as follows:

  1. The authors must supply an ORCID ID for all authors. Getting an ORCID iD is FREE, quick and easy to do through the ORCID registration page: https://orcid.org/register
  2. Abstract. What about Phe?
  3. Lines 81 – 109. The authors should not write about the results here (this should appear in the conclusions) but should state the aims of the work and what the work contains in the following (sub) chapters.
  4. Line 163. "diametric stretch marks; ---> "diametric stretch marks";
  5. MD. Some details are needed. Force field? LJ and coulomb cut-off? Delta_t? Temperature? Thermostat? Barostat? Etc.
  6. Fig. 3. (1) --> (a) and (2) ---> (b). See, Figs. 1, 2, 5 ,and 6.
  7.  
  8. Line 396, "1. New York, 1972.". 1.?
  9. Line 416. 2021, 16, 244–255. ---> 2021, 16, 244–255 (in Russian). Please add publication language (other than English) for other publications if necessary.
  10. Literature should also be standardized: the size of letters in the titles of journals as well as the titles of articles
  11. 21 self-citations [2,7,11,15-20,22,23,26-32,37-39] on 39 references 53.8% (not acceptable).
  12. In my opinion, the authors based their work too heavily on their two previous articles. The important thing is that they don't hide this problem in the article. It is up to the editor to decide what to do with this fact.

Author Response

Dear Reviewer # 3 !

Thank you very much for your comments.

All replies are in attached file. 

Round 2

Reviewer 2 Report

Authors of this work have tried to polish their paper, and have added some text as an additional interpretation. It seems that the paper is somehow improved compared to its previous version. However, the authors tried to reply that their modeling work is more important than the language they used to express their thoughts, even though they should understand that it should the opposite. A well written ms speaks a lot even with a little amount of work. In any case, this paper may be considered for possible consideration since it comprises some interesting chemical systems and results that were emerged out of their modeling. I still ask the authors to read the paper once again to ensure that all kinds of typos and grammatical errors are eliminated. 

Author Response

Dear Reviewer # 2,

Our reply is in attached file

Reviewer 3 Report

The authors have made a substantial improvement for this article. The manuscript can be accepted for publishment in the present form.

Author Response

We are very grateful to the referee # 3 for the evaluation of our work
and his recommendation to publish our manuscript.